# Familial Aggregation of a Novel Missense Variant of *COL2A1* Gene Associated with Short Extremities: Case Report and Review of the Literature

**DOI:** 10.3390/children9081229

**Published:** 2022-08-14

**Authors:** Panagiotis Christopoulos, Anna Eleftheriades, George Paltoglou, Eleni Paschalidou, Emmanouil Kalampokas, Lina Florentin, Chrysanthi Billi, Makarios Eleftheriades

**Affiliations:** 1Second Department of Obstetrics and Gynaecology, ‘Aretaieio’ Hospital, Medical School, National and Kapodistrian University of Athens, 11528 Athens, Greece; 2Postgraduate Programme “Maternal Fetal Medicine”, Medical School, National and Kapodistrian University of Athens, 11527 Athens, Greece; 3First Department of Pediatrics, ‘Aghia Sophia’ Children’s Hospital, Division of Endocrinology, Metabolism and Diabetes, Medical School, National and Kapodistrian University of Athens, 11527 Athens, Greece; 4A-Lab Genetic and Genomic Centre, 11524 Athens, Greece

**Keywords:** prenatal diagnosis, mutations, collagen, fetal limb growth abnormalities

## Abstract

We present two cases of family members (first cousins) with short extremities caused by a novel variant of *COL2A1* gene (NM_001844.5). Case 1 description: A 29-year-old woman presented in her first pregnancy for a second trimester anomaly scan at 23 weeks of gestation. Fetal long bones were measured below the third centile for gestational age. Follow-up scans revealed fetal long bone growth deceleration. Initial genetic work-up was negative and the rest of the maternal follow-up was unremarkable. A male baby weighing 3180 g was delivered at 39 weeks and 4 days of gestation. Case 2 description: A 33-year-old pregnant woman presented for a routine second trimester anomaly scan at 20 weeks and 4 days of gestation. All fetal measurements were appropriate for the gestational age. The routine growth scan performed at 32 weeks showed fetal long bone measurements below the third centile for gestational age, while the follow-up growth scan at 36 weeks and 4 days of gestation revealed consistent, below the third centile, fetal long bone growth. Given that the fetuses of these two cases were related (first cousins), whole exome sequencing (WES) was performed on Case 2. WES revealed a novel heterozygous missense variant c.1132G>A (p. Gly378Ser) of *COL2A1* gene (NM_001844.5). Subsequently, targeted genetic sequencing for the variant was performed on Case 1 and the same novel variant was found. Targeted sequencing revealed the same variant in the mother of Case 1 and the father of Case 2 (siblings). A female baby weighing 3200 g was delivered at 40 weeks and 4 days of gestation.

## 1. Introduction

Type II collagen constitutes one of the fibrillar collagens and the major component of cartilage [1]. COL2A1 mutations can cause a wide spectrum of genetic disorders, namely type II collagenopathies. These include mutations that are lethal in neonates, such as achondrogenesis type II/hypochondrogenesis, as well as nonlethal conditions; for example, Kniest dysplasia and Stickler syndrome. Genotype–phenotype studies reveal differential penetrance and expressivity within families. Features of skeletal dysplasias can be prenatally detected, guiding appropriate molecular testing [2]. The present study describes two related cases (first cousins) of fetuses showing short long bones caused by a novel variant of the *COL2A1* gene (NM_001844.5). Whole exome sequencing (WES) performed on Case 2 and targeted sequencing performed on Case 1 revealed the presence of the same novel heterozygous missense variant c.1132G>A (p. Gly378Ser) of the *COL2A1* gene (NM_001844.5), reinforcing its potential pathogenicity. Targeted sequencing revealed the same variant in the mother of Case 1 and the father of Case 2 (siblings). To the best of our knowledge, this is the first case reporting this specific variant of the COL2A1 gene prenatally. Genetic analysis revealed a correlation between phenotypical features and the c.1132G>A (p. Gly378Ser) variant, providing new sufficient information regarding its potential impact.

## 2. Clinical Report

### 2.1. Case 1

A 29-year-old nulliparous pregnant woman presented for a second trimester anomaly scan at 23 weeks of gestation. Fetal long bones were measured below the third centile for gestational age (femoral length: 35.1 mm, radius: 30.2 mm, ulna: 31.9 mm, tibia: 31.3 mm, fibula: 31.7 mm). No other structural defects were detected. Following counseling, the parents opted for genetic analysis, and an uncomplicated amniocentesis was performed. Initial genetic (array CGH—array comparative genomic hybridization) and targeted work-up for achondroplasia and hypochondroplasia were negative. Prospective third trimester growth scans showed fetal long bone deceleration (Figure 1). The rest of the maternal follow-up was unremarkable. A full-term male weighing 3180 g was delivered at 39 weeks and 4 days of gestation following an uncomplicated pregnancy. No abnormalities or complications were recorded in the neonatal history and the neonatal assessment was unremarkable. The pediatric follow-up was appropriate and there were no other defects noted. On a second uncomplicated pregnancy, a healthy baby weighing 3440 g was born at 40 weeks and 1 day of gestation.

### 2.2. Case 2

A 33-year-old pregnant woman presented in her first pregnancy for a second trimester anomaly scan at 20 weeks and 4 days of gestation. All measurements were appropriate for this gestational age. A routine third trimester growth scan was performed at 32 weeks of gestation and fetal long bones were found below the third centile for gestational age. Following counseling, the parents opted for fetal karyotyping, and uncomplicated amniocentesis was performed. The amnio PCR was negative for chromosomal abnormalities of 13, 18, 21 and X chromosomes. It was also proven that the fetus was not a carrier of the ΔF508 mutation of the cystic fibrosis gene. The fetal karyotype (both conventional and array-CGH) was normal and the targeted genetic work-up for achondroplasia and hypochondroplasia was negative. Given that the embryos of these two cases were related (first cousins), whole exome sequencing (WES) was performed on Case 2. Prenatal testing was performed in fetal amniotic fluid DNA for the presence of pathological point mutations in 1014 genes, most commonly reported to be associated with congenital fetal abnormalities and in genes associated with skeletal malformations (564 genes, see Appendix A). WES revealed a novel heterozygous variant c.1132G>A (p. Gly378Ser) of the *COL2A1* gene (NM_001844.5). A follow-up growth scan at 36 weeks and 4 days of gestation showed consistent fetal long bones growth below the third centile for gestational age (Figure 2). Apart from the abovementioned ultrasound findings, the pregnancy was otherwise unremarkable. A healthy female baby weighing 3200 g was delivered at 40 weeks and 4 days of gestation following an uncomplicated pregnancy. No other abnormalities or complications were recorded in the perinatal history. The neonatal assessment was unremarkable and there were no other defects noted. Subsequently, targeted genetic sequencing for the variant was performed on Case 1 and the same novel variant was found, namely the heterozygous variant, c.1132G>A (p. Gly378Ser) of the *COL2A1* gene (NM_001844.5). Targeted sequencing revealed the same variant in the mother of Case 1 and the father of Case 2, who are siblings (Figure 3).

## 3. Methods

Firstly, WES was performed on Case 2. Genomic DNA was isolated from an amniotic fluid sample. The applied methodology aimed to analyze and enrich the exon DNA sequences of more than 19,000 genes. Genomic DNA was captured, fragmented and enriched for exome sequences using the Whole Exome Solution by TWIST (Twist Human Core Exome Enrichment System, Twist Bioscience, South San Francisco, CA, USA). Each eluted-enriched DNA sample was then sequenced on an Illumina NextSeq-500 platform (Illumina, San Diego, CA, USA). Variant calling was performed using the VarSome Clinical platform by Saphetor. The exons of all genes that have been previously implicated in skeletal dysplasia were covered and analyzed. Each variant was evaluated based on the information from available databases, including the Human Gene Mutation Database (HGMD), ClinVar and published literature. The description of sequence variants was performed in accordance with the recommendations of the Human Genome Variant Society (HGVS). The interpretation of sequence variants was performed according to the guidelines of the American College of Medical Genetics and Genomics (ACMG). Sanger sequencing was performed to confirm the presence of the variants. WES revealed the novel heterozygous missense variant c.1132G>A (p. Gly378Ser) of *COL2A1* gene (NM_001844.5). This specific variant of the COL2A1 gene has not been detected in the GnomAD exome and genome databases and is predicted to likely be pathogenic with 13 in silico programs: BayesDel_addAF, DANN, DEOGEN2, EIGEN, FATHMM-MKL, LIST-S2, M-CAP, MPV, MutationAssesor, MutationTaster, PrimateAl, REVEL and SIFT. Sanger sequencing was performed to confirm the presence of the variant on Case 2, also on the father of Case 2, the mother of Case 1 (siblings) and Case 1. For this, primers were designed in order to amplify exon 19 of the COL2A1 gene, and targeted genetic sequencing for the COL2A1 variant was performed.

## 4. Discussion

We identified a novel heterozygous missense variant c.1132G>A (p. Gly378Ser) of the *COL2A1* gene (NM_001844.5), present in two cases who are first cousins and present with shorter long bone length. The COL2A1 gene encodes the alpha-1 chain of type II procollagen. The procollagen homotrimer (COL2A1 × 3) is formed when three alpha-1 chains fold together in a triple-helical configuration. The coding sequence region of the *COL2A1* gene includes 54 exons, and its length is 4464 bp [5]. Type II collagen constitutes the major component of cartilage and is also partly found in the vitreous gel, intervertebral discs and the inner ear. This tissue-specific expression is accountable for the clinical manifestations of short stature, ocular symptoms and hearing loss seen in the carriers of pathogenic *COL2A1* mutations [1]. The formation of the collagenous network is a complex process requiring a proper synthesis of procollagen, the generation of collagen molecules, the assembly of collagen into fibrils and, at last, the degradation of collagen fibrils [6]. Previous studies have proven that mutations seen in the triple-helical domain of a1 chains can impede the assembly, folding, intracellular transport or secretion of the type II collagens, impairing the cartilage homeostasis and, therefore, long bone development [7].

Until now, at least 460 distinct COL2A1 pathogenic mutations have been described in public databases and previous literature, causing a large spectrum of genetic disorders. In fact, these mutations were associated with at least 21 disorders [5]. The molecular spectrum of alterations includes several types of mutations, including point mutations (missense, nonsense, deletion, insertion, insertion-deletion and frame-shift mutations) and complex rearrangements [1]. The severity of the phenotype is associated with the nature of the mutation and its localization in the protein; however, no clear genotype–phenotype correlation is reported. Authors of several studies suggest that the most common mutations are missense mutations, as in our cases, leading to the substitution of glycine residue in the Gly-X-Y repeat. The mutations distributed in the Gly-X-Y triplet repeats region are often responsible for type II collagenopathies [6]. These mutations lead to the abnormal conformation and destabilization of the triple helix and, therefore, to the impairment of the proper function of type II collagen by disrupting its helical structure and result in more severe collagen type II disorders [7].

Type II collagenopathies present a wide range of severity. A wide clinical phenotypic variability and overlap in COL2A1-related disorders commonly appears in patients, even within a single family [1]. The phenotypic variability that commonly characterizes COL2A1 mutations could be explained by somatic mosaicism in some families [8]. The phenotype is also likely to be influenced by environmental factors and polymorphisms in disease-modifying genes and/or regulatory elements. Other determinants include age and environmental exposures [5]. It has been reported in the previous literature that the same COL2A1 variant could present with a different phenotype in unrelated individuals. [9]

The spectrum of type II collagenopathies includes perinatally lethal conditions, such as achondrogenesis type II (ACG2)/hypochondrogenesis, and other less severe conditions with typical onset at adolescent or adult age, such as the avascular necrosis of the femoral head (ANFH), Legg–Calvè–Perthes disease, early-onset osteoarthritis (OA) and Stickler syndrome type 1 (STL1). The nonlethal type II collagenopathies share common clinical manifestations: ophthalmic, skeletal, auditory and orofacial abnormalities [1].

Both cases presented prenatally with fetal limbs well below the third centile for gestational age and had short limbs at birth, but other abnormalities were not evident. Case 1 inherited the variant from his mother, who has a normal phenotype, whereas Case 2 inherited the variant from her father, who also has a normal phenotype. The mother of Case 1 and the father of Case 2 are siblings. Case 1 has a normal phenotype; his height is within the normal range and no other defects are noted. His growth pattern is within the normal range. We concluded that this novel heterozygous missense variant c.1132G>A (p. Gly378Ser) was responsible for the phenotype of Case 1 and Case 2. In a previous study, this variant mutation was found detected in a patient with Legg–Calvé–Perthes disease and was characterized as pathogenic [10].

Short extremities constitute a heterogeneous phenotype which is often genetic in aetiology. For many children the specific molecular causes remain unknown. The advances in exome and genome sequencing as well as in bioinformatics can assist the identification of new genetic causes of short extremities in addition to conditions such achondroplasia which have been well described [11]. As mentioned above, in this case report we identified a novel variant associated with the phenotype of short extremities. The hereditary pattern is autosomal dominant in this family, with a different penetrance and expressivity within different family members. Heterozygous pathogenic mutations in the COL2A1 gene usually follow an autosomal dominant inheritance; some cases have a de novo origin (sporadic) but rare cases of autosomal recessive inheritance have also been reported [2].

A collagen type II mutation should be suspected in fetuses or individuals with classic hallmarks of the disease, such as a short stature, spondylar, epiphyseal and metaphyseal abnormalities, skeletal dysplasia, ocular manifestations, small jaw, hearing impairment and joint hypermobility [11]. Therefore, a molecular analysis of the *COL2A1* gene should be systematically performed in those cases, since heterozygous mutations are commonly found [2]. Still, the broader spectrum offered by the simultaneous analysis of many genes using prenatal whole exome sequencing provides a valuable diagnostic tool option [5]. In cases where the molecular finding is inherited, antenatal or preimplantation genetic testing can be offered to parents due to the risk of recurrence. Family members who are asymptomatic have the option of presymptomatic testing [1].

In our bioinformatic analysis pipeline, we prefer to be able to see all variants, and, even in trio analysis, we would allow for the pipeline to see everything and evaluate ourselves using all of the available programs. This is a variant that has been characterized as pathogenic once in the ClinVar patient database and is in a position strongly conserved (phyloP100way = 7.91 is greater than 7.2). The variant has not been found in the gnomAD exomes and genomes, and has a pathogenic computational verdict based on 13 pathogenic predictions from BayesDel_addAF, CADD, DEOGEN2, EIGEN, FATHMM-MKL, LIST-S2, M-CAP, MVP, MutationAssessor, MutationTaster, Polyphen2-HVAR, PrimateAI and SIFT vs. no benign predictions. Additionally, it is positioned in the uniProt protein CO2A1_HUMAN region of interest ‘Disordered’, which has the majority of pathogenic mutations, where most of them missense. Our case presents and strongly supports exactly that: the variable expressivity of an autosomal inheritance. Despite the fact that it is inherited from phenotypically normal parents, we have two affected newborns in the same family with the same variant.

Since we are not able to predict the exact phenotype of a COL2A1 variant, prenatal counseling is of utmost importance. Previous research has indicated that COL2A1 mutations could be associated with disorders that significantly impair the quality of life of patients and their families. Children with severe skeletal dysplasia, implicated by joint pain, limb deformity or arhtritis, for example, may require a motorized wheelchair or supportive equipment. These chronic conditions can result in great psychological distress, even in social and economic issues [12,13]. Other type II collagenopathies present with a mild phenotype that progressively becomes more severe [1]. Prenatal diagnosis and counseling is necessary in such cases, since they can optimize paediatric clinical outcomes and minimize potential health risks when preventative measures are implemented in early childhood [14]. To our knowledge, 13 case reports associating the phenotype of short extremities with various COL2A1 mutations/variants have previously been reported in the literature, describing a total of 22 such cases. These are presented in Table 1 [15,16,17,18,19,20,21,22,23,24,25,26,27].

## 5. Conclusions

We identified a novel missense variant in the *COL2A1* gene in two cases (first cousins) who both had fetal limbs well below the third centile for gestational age, evident in the second trimester of pregnancy. Various disorders with a broad spectrum of phenotypes, which mainly involve abnormalities in cartilage and the ocular system, are associated with a diversity of mutations of the *COL2A1* gene. Clinical manifestations and outcomes can present great variations; previous research suggests that the same COL2A1 mutation could cause similar but different phenotypes even inside an extended family [7], something which was confirmed by our study. Prenatal diagnosis and ultrasonography have allowed for an accurate detection of skeletal abnormalities in utero, and WES can facilitate the identification of an underlying genetic cause, enabling fetal prognosis and a risk of recurrence in future pregnancies [14]. Parental counseling following prenatal ultrasound findings is a highly demanding process involving a multidisciplinary team comprising fetal medicine specialists, geneticists, neonatologists and pediatric subspecialists. An early diagnosis is crucial for the assessment and management of a potential pediatric disorder and for the provision of relevant paediatric care and follow-up. Another important aspect of early genetic diagnosis is the counseling to affected families [10]. Our study contributed to the further expansion of the COL2A1 mutation spectrum and provided more detailed information regarding their potential impact. In spite of the fact that few specific genotype–phenotype correlations have been discussed in the previous literature, the relationship of these mutations with the corresponding clinical manifestations still needs to be explored. Describing additional cases can facilitate a better understanding of COL2A1 variants in order to predict the course of the associated conditions at the early stages, leading to an improvement in childrens’ medical care and quality of life.

## Figures and Tables

**Figure 1 children-09-01229-f001:**
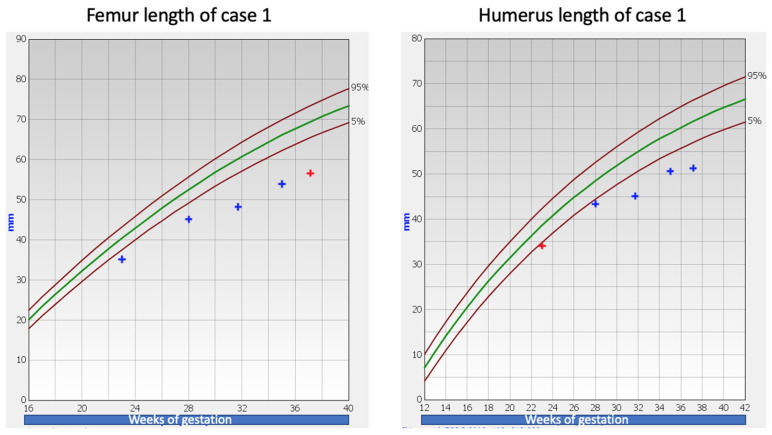
Femur and humerus length of Case 1 [3,4]. The green line represents the 50th centile.

**Figure 2 children-09-01229-f002:**
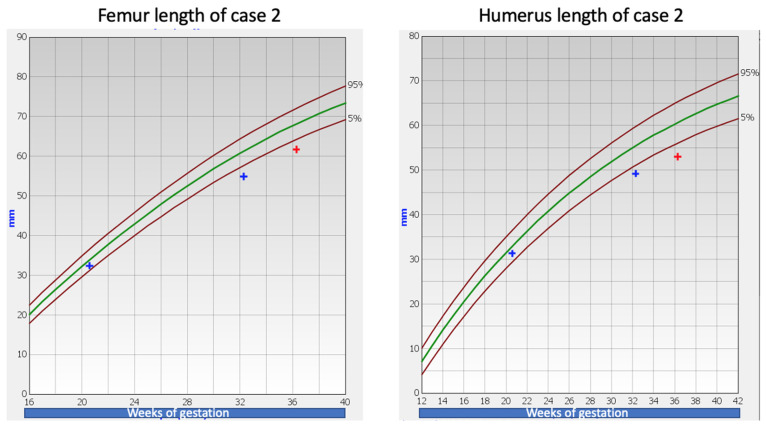
Femur and humerus length of Case 2 [3,4]. The green line represents the 50th centile.

**Figure 3 children-09-01229-f003:**
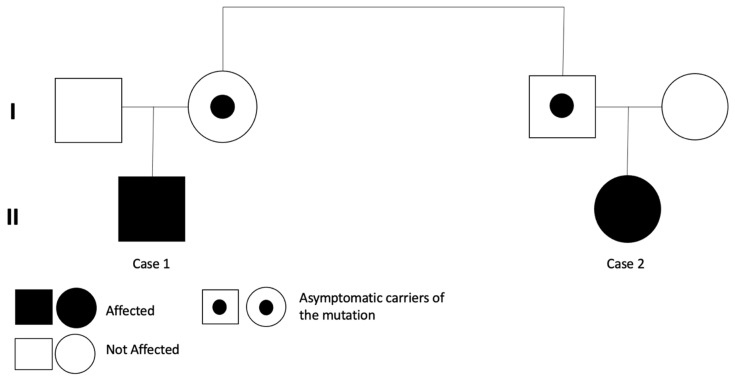
The pedigree of the family in this study.

**Table 1 children-09-01229-t001:** Existing case reports associating the phenotype of short extremities with mutations/variants of the COL2A1 gene.

Paper	Authors	Country	Year of Publication	COL2A1 Mutation	Clinical Findings	Outcomes	No. of Cases Described
1	Bedeschi et al. [15]	Italy	2011	(c.4339A > T) mutation in exon 54 resulting in a premature stop codon at amino acid 1447 (K1447X)	US examination at 17, 18 and 20 weeks revealed fetal macrocephaly, a narrow thorax, and shortening and bowing of long bones	The parents elected to continue the pregnancy. At birth the baby showed severe respiratory distress for four weeks which then resolved	1
2	Bruni et al. [16]	italy	2021	Heterozygous variant c.1267-2_1269del located in intron 20/exon 21	Achondrogenesis type II (ACG2): Brachycephaly, prominent abdomen, short limbs (at the 3rd centile), facial dysmorphism including abnormal implantation of the ears, microretrognathia, frontal bossing, a long philtrum, and congenital megalophthalmos	Termination of pregnancy at 20 weeks of gestation	1
3	Bonaventure et al. [17]	France	1995	fetus 1: heterozygous single-base mutation that substituted aspartate for glycine at position 310 of the II chain of type II procollagen. fetus 2: heterozygous single-base mutation that substituted serine for glycine at position 805	fetus 1 with achondrogenesis type II. fetus 2 with hypochondrogenesis. Both had short stature, short limbs, macrocephaly and short ribs	fetus 1 with achondrogenesis type II: termination of pregnancy at 34 weeks. fetus 2 with hypochondrogenesis: died at birth	2
4	Chung et al. [18]	Hong Kong	2012	c.2957C > T substitution in exon 41, resulting in a p.Pro986Leu mutation	At 31 years old: mid-face hypoplasia, arm-span with mild rhizomelic shortening, height below the third centile, rhizomelic shortening over both upper and lower limbs, right distal ulnar hypoplasia (Earlier radiographs taken in her childhood could not be retrieved)	The patient was born full term with a birth length below the 3rd centile. She was diagnosed with ‘‘hypochondroplasia’’ clinically in her early childhood	1
5	Desir et al. [19]	Belgium	2012	heterozygous in-frame deletion c.4458_4460delCTT (p.Phe1486del), located in exon 54	platyspondylic lethal skeletal dysplasia, Torrance type (PLSD-T): platyspondyly, extremely short limbs, and mild brachydactyly	Termination of pregnancy at pregnancy at 24 weeks of gestation	1
6	Fernandes et al. [20]	USA	1998	heterozygous G to A transition (4/8 clones) at the first position of the splice donor of intron 15, compatible with an exonskipping mutation	Kniest dysplasia: small, bell-shaped chest, short limbs (below the fifth percentile), relative macrocephaly	Born at 36 weeks of pregnancy	1
7	Forzano et al. [21]	Italy	2007	2 fetuses with the same mutation: 10370G > T missense mutation (G346V)	both fetuses (two consequtive pregnancies of a of an apparently healthy, nonconsanguineous young couple) with short extremities, micrognathia, fetal skin redundance in the neck and thorax region, and a ‘‘bell-shaped’’ thorax	fetus 1: preterm labor at 32-weeks of gestation and delivery of a 1700 g male who died in the newborn period, fetus 2: Termination of pregnancy at 18 weeks of gestation	2
8	Heinrich et al. [22]	Germany	2015	Heterozygous mutation located in 12q13.11 (c.1529G>A, p.Gly510Asp).	Prenatal sonographic examination revealed generalized hydrops fetalis and severe micromelia	Termination of pregnancy	2
9	Hochart et al. [23]	France	2015	heterozygous c.1023þ1 G>A mutation in intron 16	Kniest dysplasia: short trunk, short limbs, and midface hypoplasia	Development of chondrosarcoma/chordoma at 15 years of age	1
10	Nishimura et al. [24]	Japan	2004	Patient 1: missense mutation, c.4172A.G in exon 53. Patient 2: 4-bp deletion, c.4413-6del4 in exon 54, which predicts early termination at codon 1480	Patient 1: Prenatal ultrasonography at 29 weeks of gestation revealed polyhydramnios, thoracic hypoplasia and micromelia. Patient 2: Mid-face hypoplasia, a hypoplastic thorax, and rhizomelic shortening of the limbs were noted	Patient 1: Delivery and stillborn at 34 weeks of gestation. Patient 2: Vaginal delivery at full term. Respiratory distress soon after birth and support with intratracheal intubation which was continued from 2 days to 6 months of age. Retardation of gross motor development	2
11	Unger et al. [25]	USA	2001	two fetuses resulting from a twin prgenancy: missense mutation in exon 53. The sequence change predicted a threonine to methionine (T1370M) change in the carboxyl-terminal propeptide of type II procollagen	Short limbs were noted in both fetuses at 19 weeks gestation during ultrasound examination	Following the onset of premature labor, the twins were born by Cesarean section at 25 weeks gestation	2
12	Wada et al. [26]	Japan	2010	de novo mutation of A–C transversion in the splicing acceptor site of intron 16	Kniest dysplasia: shortening of the limbs, mild narrow thorax, and polyhydramnio	Vaginal delivery following spontaneous onset of labor at 37 weeks of gestation. Respiratory distress after birth requiring intubation, and then artificial ventilation for 10 days	1
13	Zankl et al. [27]	Switzerland	2005	Patient (fetus) 1: 4423C > T non-sense mutation in exon 52 Patient (fetus) 2: 12 bp in-frame deletion (4441_4452del, I1481_V1484del) Patient 3: missense mutation (4405G > C, D1469H) Patient 4: missense mutation (4453T > G, C1485G) Patient 5: non-sense mutation (4335G > A, W1445X)	Patient (fetus) 1: U/S at 33 weeks of gestation revealed polyhydramnion, hydrops fetalis, short extremities, bowed radius, narrow thorax and midface hypoplasia. Patient (fetus) 2: U/S revealed polyhydramnios, a small thorax with hypoplastic lungs, severe symmetrical shortening of the limbs. Patient 3: short stature at birth with short limbs, large head, narrow chest. Patient 4: disproportionate short stature at birth, severe micromelia, brachydactyly. Patient 5: short stature with short limbs and a relatively large head at birth, flat face with micrognathia, small thorax with prominent abdomen	Patient (fetus) 1: Labor was induced and the patient was stillborn at 36 weeks. Patient (fetus) 2: The child was born after 39 weeks but deceased during delivery. Patient 3: Diagnosis of Platyspondylic lethal skeletal dysplasia (PLSD) Torrance type (PLSD-T). Normal mental development in early childhood. Patient 4: Diagnosis of Platyspondylic lethal skeletal dysplasia (PLSD) Torrance type (PLSD-T). The patient required supplemental oxygen but was otherwise healthy. Patient 5: The baby died due to respiratory insufficiency	5
14	Christopoulos et al. (our case)	Greece	2022	fetus 1 and fetus 2 (first cousins): heterozygous missense variant c.1132G>A (p. Gly378Ser)	fetus 1: U/S at 23 weeks of gestation showed fetal long bones below the third centile for gestational age. fetus 2: U/S at 32 weeks of gestation showed fetal long bones below the third centile for gestational age	fetus 1: full-term male weighing 3180 g was delivered at 39 weeks and 4 days of gestation following an uncomplicated pregnancy. fetus 2: A healthy female baby weighing 3200 g was delivered at 40 weeks and 4 days of gestation following an uncomplicated pregnancy	2

## Data Availability

The authors confirm that the data supporting the findings of this study are available within the article and its Appendix A. The genetic reports are available from the corresponding author upon reasonable request.

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
