# Peer review of "Familial Aggregation of a Novel Missense Variant of COL2A1 Gene Associated with Short Extremities: Case Report and Review of the Literature"

_children, 2022, doi:10.3390/children9081229_

Round 1

Reviewer 1 Report

This manuscript described a missense mutation COL2A1( which the topic of the edited manuscript was mistaken into COLA21!). The missense mutation was found to segregated in this extended family since the two cases in this study actually were related. From the first glance this study is interesting, but if we read the abstract input in the submission system you can see it is full of typesetting errors, and the topic of the manuscript is even also erroneous, which is unbelievable.

The weakness of this study are:

  1. WES was only performed in the fetus only and the mutation they found may not be the real cause of the skeletal dysplasia since the mother of case 1 and the father of case 2 were actually phenotypically normal. It is not uncommon that variable expressivity is seen in autosomal dominant inheritance diseases, but at least the authors should discuss a bit for this specific point. In most bioinformatic analytic pipeline, such mutation may be missed or categorized as nonpathogenic if they did trio WES because the normal phenotype may be the exclusion criteria when drafting the algorithm. It is of note that this specific mutant allele is indeed considered as likely pathogenic, but the authors did not provide the necessary details of the genes they included in the analysis, which is very important and should be provided as long as revision is decided.
  2. This is Children, and there are normal phenotype family members, therefore at least a pedigree and the postnatal phenotypes/photographs should be provided or described, instead of only providing fetal phenotypes, if the authors feel this specific allele may cause skeletal dysplasia onset in utero but actually when the babies are born from outlook they did not look that dwarf, that may partly explain why the phenotypes of some family members at adults are normal.

Reviewer 2 Report

The manuscript presents two familial cases of a Novel Missense Variant of COL2 A1 2 Gene Associated with Short Extremities diagnosed prenatally, apparently for the first time. This genetic variant was diagnosed for the first case postnatally; in the third trimester of pregnancy for the second case using WES .

In my opinion, this manuscript is a short communication about the diagnosis of this variant. A case report describes a particularly challenging or unique case through an evidence-based lens with recommendations for future practice. For this manuscript, what is the research question?

I suggested :

-to discuss more prenatal counseling and prenatal management in case of the diagnosis of long bones is short, even making a chart.

-to present the postnatal consequences

- to explain why this report is important?

-to check the spelling -minor errors have occurred ( ex. Title COLA21 ; Line 169 Strickler syndrome type1)

Round 2

Reviewer 1 Report

I think they had answered my concerns. Please consider to make the genes they included in the bioinformatic pipeline process as an excel supplement, which will be of benefit to the readers.

Reviewer 2 Report

no comments
